# An Object Classification Approach for Autonomous Vehicles Using Machine Learning Techniques

Majd Alqarqaz [1], Maram Bani Younes [2,*] and Raneem Qaddoura [3]

1    Computer Science, Philadelphia University, Amman 00962, Jordan
2    Cybersecurity and Information Security, Philadelphia University, Amman 00962, Jordan
3    School of Computing and Informatics, Al Hussein Technical University, Amman 00962, Jordan
*    Correspondence: mbani047@uottawa.ca

**Abstract:** An intelligent, accurate, and powerful object detection system is required for automated driving systems to keep these vehicles aware of their surrounding objects. Thus, vehicles adapt their speed and operations to avoid crashing with the existing objects and follow the driving rules around the existence of emergency vehicles and installed traffic signs. The objects considered in this work are summarized by regular vehicles, big trucks, emergency vehicles, pedestrians, bicycles, traffic lights, and traffic signs on the roadside. Autonomous vehicles are equipped with high-quality sensors and cameras, LiDAR, radars, and GPS tracking systems that help to detect existing objects, identify them, and determine their exact locations. However, these tools are costly and require regular maintenance. This work aims to develop an intelligent object classification mechanism for autonomous vehicles. The proposed mechanism uses machine learning technology to predict the existence of investigated objects over the road network early. We use different datasets to evaluate the performance of the proposed mechanism. Accuracy, Precision, F1-Score, G-Mean, and Recall are the measures considered in the experiments. Moreover, the proposed object classification mechanism is compared to other selected previous techniques in this field. The results show that grouping the dataset based on their mobility nature before applying the classification task improved the results for most of the algorithms, especially for vehicle detection.

**Keywords:** autonomous vehicle; object detection; object classification; Udacity dataset; BDD100K dataset; machine learning; road network

## 1. Introduction

Autonomous vehicles or self-driving cars have been proposed recently over the road network. They are also known as driverless vehicles that can operate and execute vital activities without the assistance of humans [1]. Thus, these vehicles should be able to investigate their surrounding environment accurately. Several systems have been developed to improve the performance of autonomous vehicles in several road scenarios and situations. Some systems have been designed to drive under bad weather conditions (i.e., fog, rain, snow) [2–4]. Other systems have been developed to clearly and accurately control the vehicles at curving roads, junctions, traffic lights, blind turns, and on-street parking [5–10].

On the other hand, many researchers have worked on object detection and classification problems over the road network (i.e., regular vehicles, emergency vehicles, cyclists, trucks, pedestrians, etc.) [11–15]. These studies effectively aim to reduce the increasing number of traffic accidents over road networks. They also reduce the driver's stress and tension during their trips [16]. More importantly, they aim to help ultimately replace the drivers in autonomous vehicle scenarios. The surrounding environment of autonomous vehicles is essential for safe road network trips. Investigating the existing objects should be accurate and quick to avoid crashes and accidents. The system of autonomous vehicles is based mainly on perception and decision-making processes. For example, the route

of autonomous vehicles toward the targeted destinations should be set based on some predefined parameters. This includes the physical location of the destination, the map of the surrounding road network, and the navigation trip on the road network [17,18].

Quick responses and actions are required from autonomous vehicles upon detecting any object over the road network. These responses depend mainly on the main characteristics of the detected object. Determining the main features of the existing objects: their nature, location, and size is one of the most critical problems that require further development in the system of autonomous vehicles. For instance, vehicles have to decrease their speed to increase the in-between safe distances if a nearby vehicle is detected traveling in front of them. They must change their traveling lane and open the way for emergency vehicles if they are seen behind them. Moreover, warning signs on the road should be analyzed and understood to react accordingly by following the recommended speed or noticing the existing exit point. Detecting a located traffic light at a signalized road intersection determines whether that vehicle can pass through that intersection or it should wait for the green signal.

Machine learning (ML) techniques, such as deep convolution neural networks, have been used in computer vision to detect objects over the road network. The object detection problem over the road network has been handled by analyzing the road's closed-circuit television (CCTV) footage. With the help of a CCTV camera, images are taken every second. Thus, every vehicle on the road is detected in every image. These studies have classified objects after detecting them [19]. Processing images acquired robust light detection and extended-range equipment. Moreover, some research studies have focused on the correlated problem of vehicle sensor location problem [20–25]. Besides, LiDAR and radar technologies can generate a map of its environment to detect, locate, and track moving targets [26]. Vehicles, pedestrians, bicycles, motorcycles, and other obstacles on or beside roads are all objects of interest in automotive driving applications [26]. Thus, radars provide direct perception-related inputs by extracting depth-related characteristics with highly predictable processing approaches. Then, RGB cameras are used to create images that replicate human vision, capturing light in red, green, and blue wavelengths. Processing these images can help to identify existing objects, differentiate them from the background, and analyze a scene. On the other hand, the Global Positioning System (GPS) helps in the navigation system of these vehicles by determining the longitude, latitude, speed, and direction of each vehicle.

In this work, we aim to use data classification techniques as an intelligent object classification approach. This is to classify the detected objects according to their main characteristics. We have investigated the performance of these techniques based on several datasets. Then, collecting several considered objects into three main groups based on the mobility nature of these objects (i.e., vehicles, people, and signs) is applied. The the grouped datasets obtained better Accuracy, Precision, F1-Score, G-Mean, and Recall results. We compare the performance of six main machine learning algorithms to recommend the most suitable one for object detection on road networks. The performance of the proposed approach has been improved by modifying the datasets and grouping similar objects in a single group.

The rest of this paper is organized as follows: Section 2 investigates some recent relevant studies about object detection and classification in autonomous driving scenarios. Section 3 provides a brief discussion of the classification algorithms utilized in this work. Section 4 presents the pre-processing techniques we applied to prepare the dataset. In Section 5, the experiments and results are presented, discussed, and compared to previous studies. Finally, Section 6 concludes the entire paper.

## 2. Literature Review

Object classification is a method of determining which class instances an object belongs to. Autonomous vehicles must classify objects over the road network to take different actions with different existing objects. Moreover, the location of each detected object should be precisely determined. To obtain a complete 3D perspective of the area, object detection

is becoming a subdomain of computer vision (CV) [27]. Among the goals of self-driving cars include saving lives and increasing safety by minimizing accidents, as well as making private transportation possible, efficient, and reliable [28,29]. Specific object recognition is a sub-problem of the general object recognition problem. This requires assigning distinguishing attributes to each object with proper names [30]. Three-dimensional object detection is a crucial task for the autonomous driving of an optical navigation module. Various sensors, such as millimeter-wave radar, camera, or laser radar (LiDAR), provide road scene information to the optical navigation module. Then, classification techniques process the gathered information to give a derivable area for autonomous vehicles [31]. Udacity presented the dataset generated from the GTI (Grupo de Tratamiento de Imágenes, Madrid, Spain) vehicle image collection and the KITTI (Karlsruhe Institute of Technology, Karlsruhe, Germany and Toyota Technological Institute, Nagoya, Japan) vision benchmark suite. It employs a histogram of oriented gradients (HOG) feature extraction algorithm to recognize multiple vehicles in images and classify them using various classification techniques. The experimental result shows that the efficiency is higher while using the LR when compared to the decision tree (DT) algorithm and the support vector machine (SVM) [32].

Ortiz Castelló et al. have evaluated version 3 of the "You Only Look Once" (YOLOv3) and YOLOv4 networks by training them on a large, recent, on-road image large-scale Berkeley Deep Drive (BDD100K) dataset, with a significant improvement in detection quality. Additionally, some models were retrained by replacing the original Leaky Rectified Linear Unit (Leaky ReLU) convolution activation functions from the original YOLO implementation with two advanced activation functions. The self-regularized non-monotonic function (MISH) and its self-gated counterpart (SWISH) resulted in significant improvements in detection performance over the original activation function. YOLO is a real-time object detection algorithm that identifies specific objects in videos or images. YOLO uses features learned by a deep convolutional neural network to classify each object. The BDD100K dataset was used to train the algorithms. It is a large, comprehensive dataset that includes a variety of objects in various weather conditions, locations, and times of day, as well as a wide range of light conditions and occlusion. Average Precision (AP) is the primary measure used in this comparison study. The MISH model gets the best performance, followed by the SWISH function. However, both lead to better results than the original Leaky ReLU implementation [27].

Moreover, deep convolutional networks are used to classify objects accurately by Mobahi and Sadati [29]. This study used the BDD100K dataset to train and test the algorithm using Python and the open-source PyTorch platform using the CUDA tool, which allows image processing. The experiments were performed using a single-shot multi-box detector (SSD), faster R-CNN (Region-Based Convolutional Neural Network), and PyTorch algorithms. They classified three scales of objects: small, medium, and large [29]. Karlsruhe Institute of Technology, Toyota Technological Institute (KITTI), and Multifog KITTI datasets have been used in other experimental studies. Mainly, the AP measure has been determined to compute the performance of the 3D object detection task. The findings were greatly enhanced by employing a Spare LiDAR Stereo Fusion Network (SLS-Fusion) [2]. Then, the proposed 3D object detection algorithm divides objects into three difficulty levels: easy, moderate, and hard. Based on the 2D bounding box sizes, occlusion, and truncation extents appearing on the image. The hard level focuses on classifying objects in foggy weather for self-driving vehicles [2].

Furthermore, Mirza et al. used YOLOv3, PointPillars, and AVODS (Aggregate View Object Detection) methods to detect and classify objects over the road network. These methods perform much better on the KITTI dataset than the NuScenes, Way Forward in Mobile (Waymo, Mountain View, CA, USA), and A*3D datasets. On night scenes, the mean Average Precision (mAP) achieved by PointPillars is the best. However, it fails in adverse weather situations such as rain [4]. Then, to increase the performance of classifying objects in foggy weather circumstances, Mai et al. trained the Spare LiDAR Stereo Fusion Network (SLS-Fusion) using the KITTI dataset (i.e., the Multifog KITTI dataset). In addition, Al-Rifai

has used the YOLOv3 algorithm with Darknet-53 CNN for object classification on the road network; they detect and classify cars, trucks, pedestrians, and cyclists [13]. On the other hand, Krizhevsky et al. [33] proposed a deep convolutional neural network that has been used to extract the image representations automatically. The perception system of an autonomous vehicle converts sensory data into semantic information, such as lane marking, drivable areas, and traffic sign information. Moreover, this system has been developed to identify and recognize objects on the road (e.g., vehicles, pedestrians, cyclists) [34]. Cameras can recognize pedestrians using a convolutional neural network (CNN) and determine vehicle positions by merging the picture position with the LiDAR point cloud [35]. Deep learning can be used to process the sounds of emergency vehicles from a long distance and determine the direction from which they are approaching. Autonomous vehicles must notice the responding emergency vehicles [36].

On real and synthetic data, 3D object detection and classification are implemented by Agafonov and Yumaganov [37]. Vehicles are used as detected objects, the KITTI dataset is used, and the open-source simulator Consul Auditing and Reporting Language (CARLA) is used as a source of synthetic data. This simulator is provided for testing diverse traffic situations to train autonomous vehicle control systems. The AP of classification objects is measured for these scenarios as an evaluation measure [37]. Table 1 summarizes the recent studies in object classification mechanisms over the road network, illustrating their main findings and limitations. After looking over the most recent and pertinent studies, we may summarize the deficiencies and weaknesses as follows:

- Some investigations produced results that showed a considerable increase in the amount of time needed for processing.
- Some studies fail in adverse weather situations such as rain.
- They did not perform well in the classification of large-scale objects.
- When it comes to categorizing large-scale items, several of the earlier studies did not perform particularly well.
- Some studies have a high misclassification rate for small objects compared to larger ones.
- There is still room for improving the accuracy measures achieved by the current studies, which results in a higher overall quality of the findings.

**Table 1.** Summary of the previous research studies.

| Paper | Tested Algorithms | Main Objective | Finding | Measures | Limitation | Dataset |
|---|---|---|---|---|---|---|
| (Ortiz Castell´o et al., 2020) [27] | YOLOv3, YOLOv4, MISH, and SWISH | Obtain more accurate models to confidently classify the on-road conditions aimed to decrease limitations in processing capacity and bandwidth | YOLOv4 convolutional layers using the new MISH and SWISH functions produced better results with minor improvements in classification quality; MISH was the function that made the best results | Average, Precision, and AP5 | The expense of an essential rise in processing time | MS-COCO and BDD100K |
| (Mobahi and Sadati, 2020) [29] | SSD, Faster R-CNN, and PyTorch | To improve the classification in a simple way of different scales, especially small ones from the self-driving car | Compared to recent approaches, it performed better in the category of small-scale objects by PyTorch and improves the accuracy of object classification at various scales, especially the small ones | Average, Precision, (IoU), Average Precision (scale) | Detecting large-scale objects | BDD100K |
| (Mai et al., 2021) [2] | SLS-Fusion, Pseudo-LiDAR++ | Investigate the effects of fog on object classification in driving scenarios to increase performance in foggy weather | This result is very satisfying because it indicates the SLS-Fusion algorithm's robustness while dealing with foggy datasets | Average Precision (AP) over union (IoU) thresholds at 0.5 and 0.7 | Depends on the quality of cameras or LiDAR used | KITTI Clear +MultiFog |

**Table 1.** *Cont.*

| Paper | Tested Algorithms | Main Objective | Finding | Measures | Limitation | Dataset |
|---|---|---|---|---|---|---|
| (Mirza et al., 2021) [4] | YOLOv3, PointPillars, AVODS | Aim to handle autonomous object classification systems in weather circumstances including rain, fog, and snow, and examine how performance degrades when weather conditions degrade | Detectors that work well in clear weather can fail in adverse weather conditions; to cover such situations benchmarking datasets must be improved | Average Precision (AP50) | Fails in adverse weather conditions such as rain | NuScenes, Waymo, KITTI, A*3D |
| (Al-refai and Al-refai, 2020) [13] | YOLOv3-Darknet CNN | The goal is to detect four different categories of pedestrians, vehicles, trucks, and cyclists, and to evaluate dataset images gathered from public roadways using the front-facing camera of a vehicle | YOLOv3-Darknet algorithm has high misclassification rate for small objects like pedestrians and cyclists compared to larger objects like cars | Precision Recall | High rate of misclassification for small objects compared to larger objects | KITTI |
| (Agafonov and Yumaganov, 2020) [37] | AVOD, PointRCNN, and SECOND | The object classification experiment was on a car category to evaluate the performance of the algorithms for 3D object detection and classification on both real and synthetic data | 3D objects detection and classification methods when trained on synthetic data cannot be applied to detect objects with real data | AP on 2D images, AP on BEV projections, AP of 3D objects | Fails to detect 3D objects in realistic synthetic data | KITTI CARLA |

This research seeks to design an intelligent object classification approach for autonomous vehicles, as well as provide an efficient model to address these weaknesses.

## 3. Background

This section briefly discusses some specialized methods and algorithms tested and utilized in this work. This includes decision tree (DT), naive Bayes (NB), K-nearest neighbor (KNN), stochastic gradient descent (SGD), multi-layer perceptron (MLP), and logistic regression (LR) algorithms.

### 3.1. Decision Tree (DT)

One of the earliest and most well-known ML algorithms is the DT. DT is an ML algorithm that divides learning activities. The tree is constructed by dividing the dataset into smaller sets until each partition is clean and pure. In a typical DT, each node will exist on multiple levels, and the node at the very top of the tree, known as the root node, will be the first level. Every internal node is a test on one of the input variables or attributes. Depending on the test result, the classification algorithm takes a branch to the right or left to reach the appropriate child node. The testing and branching processes are repeated until it reaches the leaf node. The leaf nodes, also known as terminal nodes, correspond to the decisions made [38,39]. This structure of the DT algorithm is illustrated in Figure 1.

### 3.2. Naive Bayes (NB)

Naive Bayes (NB) is a method of classification that is founded on Bayes' theorem. This theorem introduces the concept of feature condition independence, and the classification model is simple and clear [40]. Equation (1) below illustrates the considered model.

$$P(A/B) = \frac{P(A) \times P(B/A)}{P(B)} \tag{1}$$

Given that *B* has already occurred, we can calculate the likelihood of *A* occurring in the future. In this case, *B* is the evidence, while *A* is the hypothesis. Independent predictors and characteristics are assumed in this example. In other words, the presence of a given trait does not affect the presence of another [41].

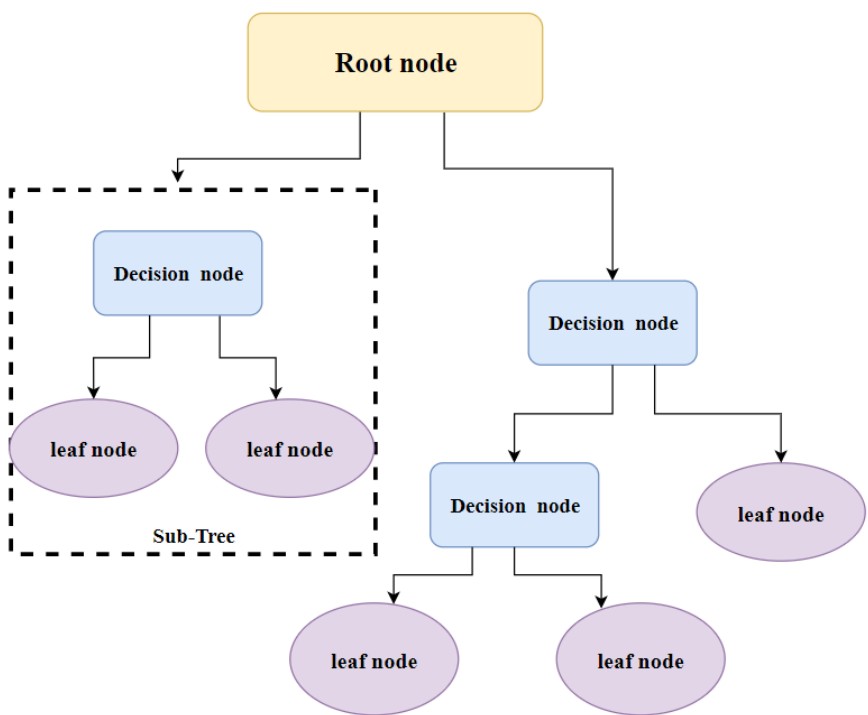

**Figure 1.** DT algorithm structure.

### 3.3. K-Nearest Neighbor (KNN)

The K-nearest neighbor (KNN) algorithm is a supervised learning method. This approach depends on measuring the distance between the newly input data that needs to be classified and all of the other data in the dataset. The 'K' in the KNN algorithm refers to the number of nearest neighbors considered when casting a vote. The same sample item can be classified in various ways depending on the value one chooses to assign to the 'K' variable. Finding a record of K-nearest neighbors involves discovering all most similar to it in terms of standard features. This stage is also known as distance calculation or similarity search. The distance can be calculated using several equations such as Euclidean distance (*ED*), Manhattan distance (*MD*), and others [42]. Equation (2) illustrates how to compute the Euclidean distance. Equation (3) computes the Manhattan distance between any two objects (i.e., X, Y).

$$ED : d(x,y) = \sum_{i=1}^{n} (x_i - y_i)^2 \qquad (2)$$

$$MD : d(x,y) = \sum_{i=1}^{n} |(x_i - y_i)| \qquad (3)$$

For classification issues, the algorithm assigns a class label based on the majority vote (i.e., the label that appears more frequently in neighbors). The accuracy of the findings is determined by comparing the model's predictions and estimations to the available classes in the testing set [39,41]. Figure 2 illustrates the KNN algorithm where the arrows point to the K nearest nodes to the target node.

### 3.4. Stochastic Gradient Descent (SGD)

This algorithm is called stochastic or minibatch gradient descent because, each time it iterates, it chooses new samples randomly to put in the minibatch. However, this algorithm needs to figure out an extra parameter: the size of the minibatch it uses. It is a widely used and popular algorithm as the ML algorithm [43]. Stochastic refers to a method or technique associated with a random chance. Consequently, a few random samples are selected instead of the whole dataset for each iteration. SGD seeks to discover the global minimum by

altering the network's structure after each training step. This method decreases error by estimating the gradient for a randomly selected subset rather than determining the gradient for the entire dataset. In practice, random sampling entails shuffling the dataset randomly and proceeding through the batches sequentially. SGD conducts frequent high-variance changes that permit significant variations in the objective function [44].

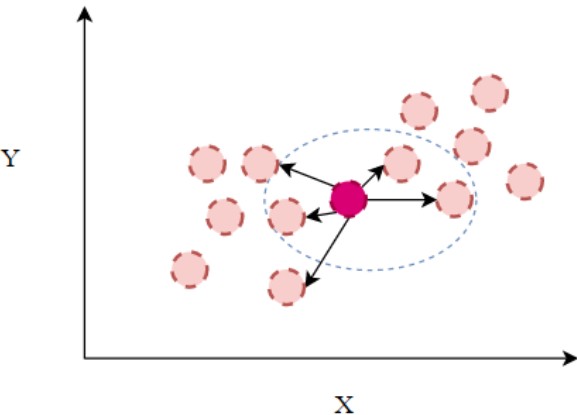

**Figure 2.** KNN algorithm structure.

*3.5. Multi-Layer Perceptron (MLP)*

A multi-layer perceptron is a supervised learning algorithm that trains functions through a dataset. MLP is structured similarly to how the brain evaluates and processes information. The three primary layers are the input layer, at least one hidden layer, and the output layer. The values of the features are sent to the input layer. The hidden layer is positioned between the input and output layers, and the values of the artificial neuron nodes are calculated by multiplying the total number of input node values by their weights. This is illustrated in Equation (4).

$$\sum_{i=1}^{n}(x_i w_i) \tag{4}$$

where $n$ is the number of input nodes, $x$ is the value of an input node $i$, and $w$ is the weight of the input node $i$. The weights determine how much impact the input has on the output. The value of each neuron is then generated using an activation function. The activation function maps any real input into a confined range, commonly [0, +1] or [−1, +1]. Finally, the value of the neuron at the output layer is calculated by multiplying the total number of the hidden layer neuron values with their assigned weights, which determines the predicted output.

MLP and feed-forward neural networks (FNN) are two names for the same type of deep neural network. The perceptron is a well-known ML technique that inspired neural networks [45]. The three layers architecture is illustrated in Figure 3.

**Input Layer**   **Hidden Layer**   **Output Layer**

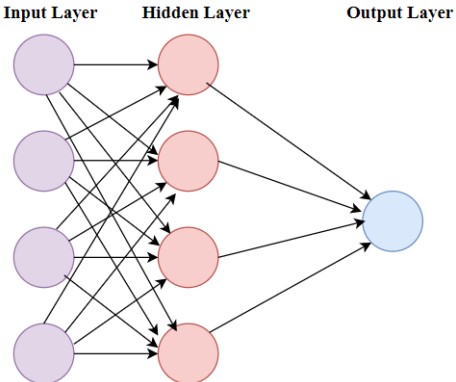

**Figure 3.** MLP algorithm structure.

*3.6. Logistic Regression (LR)*

Logistic regression is a statistical classification model that counts the association between a categorical dependent variable and one or more independent variables. This is by utilizing probability scores as the predicted values of the dependent variable [46].

## 4. The Proposed Object Detection Approach for Autonomous Vehicles

In this section, we start by discussing the methodology of the proposed approach. Figure 4 illustrates the steps of the proposed approach methodology. The Udacity and BDD100K datasets are collected from Roboflow and Kaggle repositories. Data preprocessing techniques are applied to handle the missing values in the datasets. In addition, the datasets are split into 80% training data and 20% testing data. The most popular ML mechanisms, such as the DT, NB, KNN, SGD, MLP, and LR algorithms, are used to classify the objects for the Udacity and BDD100K datasets. Finally, the five main evaluation measures for evaluating a classification model are Accuracy, Precision, F1-Score, G-Mean, and Recall. In the second stage, we divided the BDD100K dataset into groups, and the steps mentioned above were re-applied to these groups to obtain a new model compared to the model produced from the first stage.

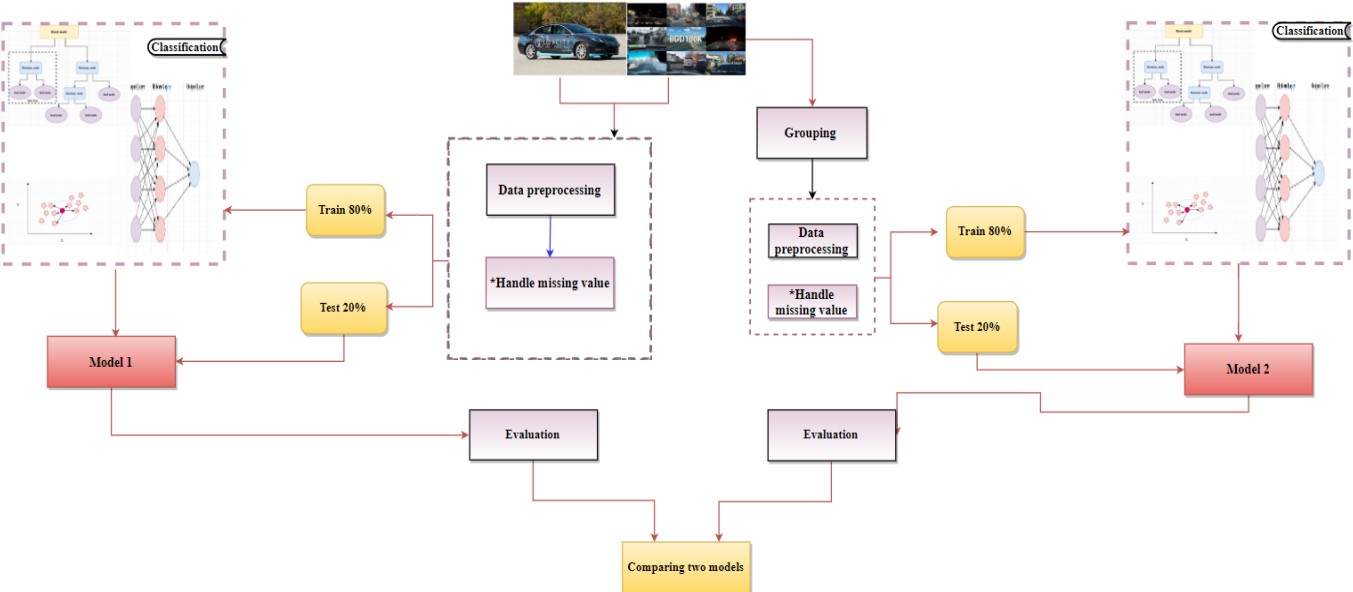

**Figure 4.** The methodology model of the proposed approach.

We provide the procedure followed in preparing and cleaning the used datasets. The imbalanced data issue is discussed as its effects on the obtained results. Making the data more complete and getting more accurate results can be achieved by preprocessing the data. The data preparation process comprises executing Python code to find missing and duplicated values. Every class has a numeric value in the Udacity and BDD100K datasets. The data preprocessing techniques are applied to handle the missing values in the Udacity and BDD100K datasets. Then, the datasets are split into 80% as training and 20% as testing data. The classification algorithms are applied to classify the dataset. Finally, the measures for the algorithms are calculated. The main categories in the Udacity dataset include pedestrian, car, truck, traffic light, and biker. In contrast, the BDD100K dataset includes pedestrian, rider, car, truck, bus, motorcycle, bicycle, traffic light, and traffic sign categories. We can see that the BDD100K is much more sophisticated compared to the Udacity dataset.

*4.1. Data Collection*

The Udacity dataset was collected using Udacity's self-driving car simulator [47]. With this simulator's training mode, drivers can record themselves driving the car on certain

tracks. This method is called "behavioral cloning" because it copies the user's actual actions as they navigate the vehicle. The information is shown as images taken from three angles, mainly from the center, left, and right. The dataset contains 97,942 labels across 11 classes and 15,000 images, including thousands of pedestrians, bikers, cars, and traffic lights. This dataset was exported via Roboflow [48].

On the other hand, the BDD100K dataset is one of the most commonly used datasets for object detection and classification in autonomous driving. The diversity of the data is essential to test the robustness of perception algorithms. This dataset includes a wide range of scene types, including city streets, residential areas, and highways. In addition, the videos were taken in a wide range of climatic conditions and at various times throughout the day. In this dataset, there are up to 90 objects per image [29]. The BDD100K dataset contains 13 classes (i.e., traffic sign, traffic light, pedestrian, rider, car, bus, truck, train, motorcycle, bicycle, vehicle, another person, and trailer). General snapshots consisting of several selected images taken from the Udacity and BDD100K datasets are shown in Figures 5 and 6, respectively.

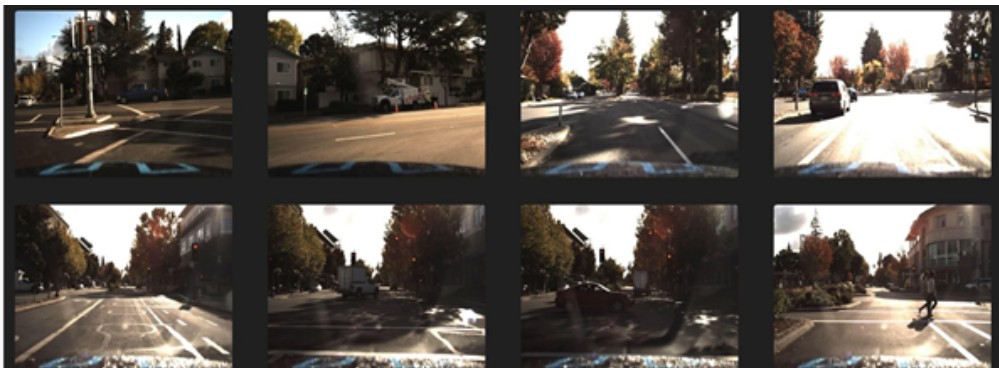

**Figure 5.** A snapshot of several images from the Udacity dataset.

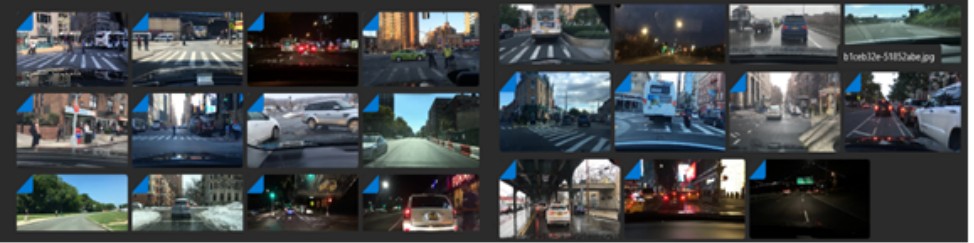

**Figure 6.** A snapshot of several images from the BDD100K dataset.

*4.2. Data Preparation*

We cleaned the data for more accurate results in the Udacity dataset. This cleaning was by combining all of the classes that fall under the umbrella of the traffic light categories into a single class that we have named "Traffic Light". The combined group comprises trafficLight-GreenLeft, trafficLight-Green, trafficLight-RedLeft, trafficLight-Yellow, trafficLight-YellowLeft, and trafficLight-Red. Then, 80% of the dataset is used for training, and 20% is used for testing. Moreover, data preprocessing techniques are applied to handle the missing values in the dataset, which involves executing code to discover missing and duplicated values [47,48].

On the other hand, to clean the data and obtain more accurate results, classes that did not record high views were eliminated to enhance the data balance at the BDD100K dataset. The eliminated types include other persons, vehicles, trains, and trailers, rarely detected over the road networks in real scenarios. Data preprocessing techniques are applied to handle the missing values in the dataset, a preparation that involves executing code to

discover missing and duplicated values. In addition, the dataset offers real-world images while working within constrained environments and attempting challenging tasks.

### 4.3. Grouping Modification

The BDD100k dataset is imbalanced data, and the performance of the tested classification algorithms is low in terms of Accuracy, Precision, F1-Score, and G-Mean on this dataset compared to the Udacity dataset. To enhance the balance situation of the tested dataset, the instances of the BDD100K dataset were grouped based on the nature of the objects. This should improve its performance. These groups were obtained by gathering objects of an exact nature together. These groups include vehicles, people, and signs collected from the nine classes that appeared in the previous section. The pedestrian and rider objects are grouped in the people category. The traffic light and traffic sign objects are grouped in the sign group. Finally, the car, truck, bus, motorcycle, and bicycle objects are all grouped in the vehicle group.

## 5. Experiments and Results

In this section, we first present the tested experiments on the investigated datasets. Then, we discuss and analyze the obtained results. Five main evaluation measures have been used in these experiments: Accuracy, Precision, F1-Score, G-Mean, and Recall. The accuracy of a classifier is measured as the proportion of the total number of correct predictions. Precision measures the number of instances accurately recognized as positive relative to all the optimistic predictions, whereas Recall measures them relative to all the positive instances. Moreover, F1-Score is a weighted average of Precision and Recall. Finally, G-Mean evaluates classification results on both the majority and minority classes equally. Even if the negative instances are classified correctly, a low G-Mean indicates poor performance in classifying the positive cases [49]. G-Mean maximizes each class's accuracy while keeping this accuracy balanced [50]. These measures range from 0 to 1, where 1 means the highest score. These measures are used to evaluate and compare the performance of six main classification algorithms. The DT, NB, KNN, SGD, MLP, and algorithms are considered in this experimental study.

Object classification is a computer vision task that classifies the visual objects gathered in digital pictures from photos and video frames into different classes, such as persons, traffic lights, vehicles, and bicycles. The Udacity and BDD100K datasets are the most commonly used for object classification in autonomous driving environments. This section shows the results of classifying the objects over the road network for autonomous vehicles to investigate their surrounding environment on these datasets. Moreover, it compares the obtained results of our approach to previous studies in this field.

### 5.1. Evaluation of Classification Algorithms on the Udacity Dataset

Table 2 illustrates the results obtained using the Udacity dataset for the main investigated classification algorithms. As observed from the results, the DT algorithm produced the best results when it was used to train a model. It achieves a G-Mean value of 98%, representing an excellent percentage of positive predictive values concerning all the predictive values. However, the results of the rest of the measures equal a percentage of 97%. Compared to the DT, the G-Mean of the KNN algorithm is 2% lower. We also note that the MLP comes third with good results after KNN; it has a 1% lower G-Mean value than the KNN with a value of 95% for the G-Mean measure. The MLP algorithm is followed by the SGD algorithm, which achieved 90% G-Mean, 5% lower than the MLP. The LR is 2% lower than the SGD; its score is 88% in G-Mean. Comparatively, the NB has the lowest score of all of the algorithms with 87% G-Mean and it achieves 80% at the rest of the measures.

**Table 2.** Score values of the various measures for each algorithm in the (Udacity) dataset.

| Algorithms | Accuracy | Precision | F1-Score | G-Mean | Recall |
|:---:|:---:|:---:|:---:|:---:|:---:|
| DT | 97% | 97% | 97% | 98% | 97% |
| NB | 80% | 80% | 80% | 87% | 80% |
| KNN | 94% | 94% | 94% | 96% | 94% |
| MLP | 91% | 91% | 91% | 95% | 91% |
| SGD | 84% | 84% | 84% | 90% | 84% |
| LR | 81% | 81% | 81% | 88% | 81% |

Figure 7 shows a graphical representation of these measured measures: Accuracy, Precision, F1-Score, G-Mean, and Recall of all the algorithms for the Udacity dataset. It should be clearly noted from the figure that the DT algorithm is the best algorithm that obtained the highest accuracy. We can also rank the algorithms according to their performance for all the measures as DT, KNN, MLP, SGD, LR, and NB, where DT has the highest and NB has the lowest values.

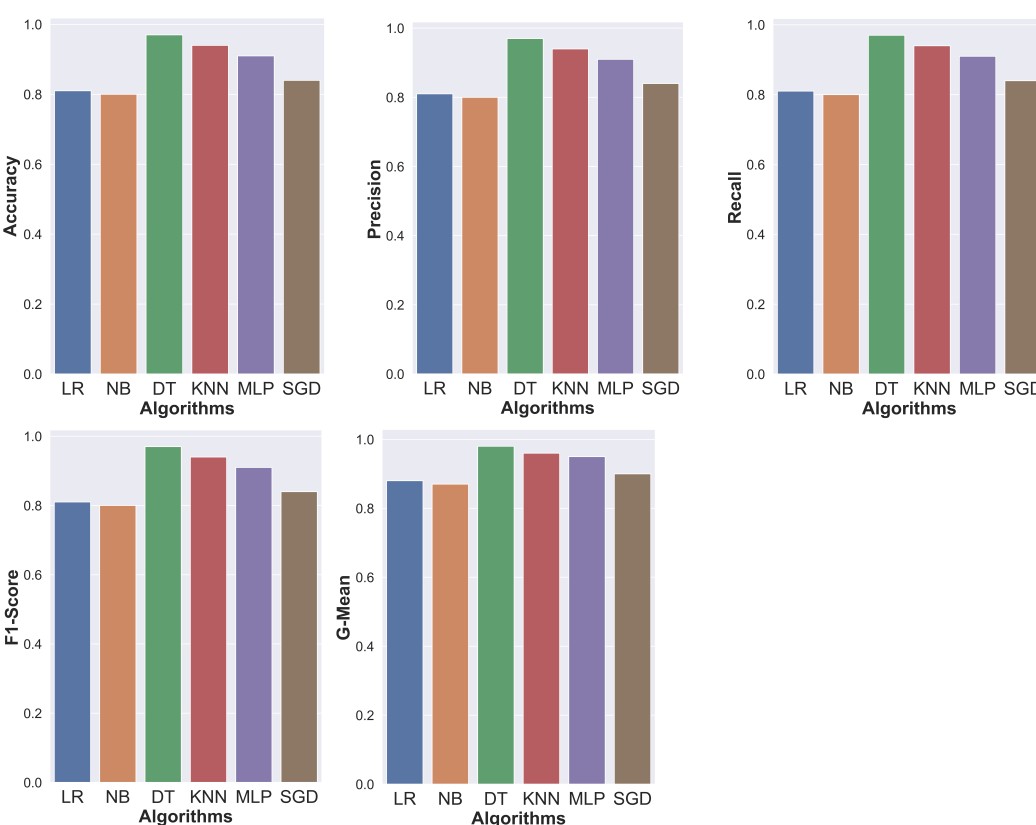

**Figure 7.** Score values of the various measures for Udacity dataset.

### 5.2. Evaluation of Classification Algorithms on the BDD100K Dataset

The results we obtained using the BDD100K dataset are presented in Table 3. MLP achieved the best results, with a G-Mean value of 84% (i.e., the best result among all measures for the same algorithm). This measure considers the balance between the classification performances of both the majority and minority classes. The results of the other measures are all equal, with a percentage of 72%. The KNN algorithm's G-Mean measure is 2% lower than the MLP. Compared to other algorithms, the SGD and DT algorithms have the lowest G-Mean score of 75%. In contrast, the DT algorithm outperforms the SGD algorithm in all other measures by 1%, with an overall score of 59%.

**Table 3.** Score values of the various measures for each algorithm for the BDD100K dataset.

| Algorithms | Accuracy | Precision | F1-Score | G-Mean | Recall |
|:---:|:---:|:---:|:---:|:---:|:---:|
| DT | 59% | 59% | 59% | 75% | 59% |
| NB | 59% | 59% | 59% | 76% | 59% |
| KNN | 68% | 68% | 68% | 82% | 68% |
| MLP | 72% | 72% | 72% | 84% | 72% |
| SGD | 58% | 58% | 58% | 75% | 58% |
| LR | 62% | 62% | 62% | 78% | 62% |

Figure 8 graphically shows the score measures: Accuracy, Precision, F1-Score, G-Mean, and Recall of the tested algorithms for the BDD100K dataset. From the figure, it should be noted that the MLP Algorithm is the best that obtained the highest Accuracy. KNN followed by LR has the best G-Mean, which achieved 78% G-Mean, 4% lower than the KNN and 6% lower at the rest of the measures with a percentage of 62%. While NB is 2% lower than the LR, its score was 76% in the G-Mean and is 3% lower at the rest of the measures with a percentage of 59%.

### 5.3. Evaluation of the Grouped Dataset

In this section, we evaluate the same measures on the grouped BDD100K dataset, displayed in Table 4. The results here are much higher than those on the same ungrouped dataset. The LR, NB, KNN, and MLP are the best algorithms that obtained the highest score across all of the measures for the vehicle group. They scored 96% for G-Mean. These algorithms are followed by the SGD algorithm, which achieved 95% G-Mean, 1% lower than these algorithms. Comparatively, the DT algorithm has the lowest score of all of the algorithms, 2% lower than the SGD; its score was 93% in G-Mean. The other measures recorded a value of 94% for LR, NB, KNN, and MLP algorithms, a value of 89% for the DT, and a value of 93% for the SGD algorithm.

For the people group, the LR, NB, KNN, and MLP are the best algorithms that obtained the highest score across all measures. For the sign group, the MLP is the best algorithm that obtained the highest score across all of the measures; it scored 78% across all. The MLP was followed by the LR, which was 1% lower than the MLP algorithm. KNN is 5% lower than the MLP, its score was 73% across all of the measures.

Figure 9 illustrates score measures: Accuracy, Precision, F1-Score, G-Mean, and Recall of the tested algorithms for the obtained groups based on the nature (i.e., vehicles, people, and signs) from the BDD100K dataset. It is observed from the figure that the vehicle group gets the highest score in all measures. The results of the signs group were the worst among them.

### 5.4. Comparison Results

This section compares our obtained results to another recent comparable paper. It also compares the score for the BDD100k dataset overall classes with the scores from the modified grouped dataset.

The DT algorithm in our approach produced an Accuracy of 97%, while C. R. Kumar [32] obtained 93%. We have also seen an increase in the Precision measure; we obtained 98% while they obtained 90%. Furthermore, the F1-Score and Recall were 97%, whereas the score for the other paper was 90%. We have achieved a higher percentage in Accuracy, Precision, F1-Score, and Recall measures compared to C. R. Kumar [32]. This is justified by the fact that, when we cleaned the dataset, we combined all of the classes that fall under the umbrella of the traffic light categories into a single class that we have named "Traffic Light". Table 5 shows the comparison results between recent similar work and our contribution to the Udacity dataset.

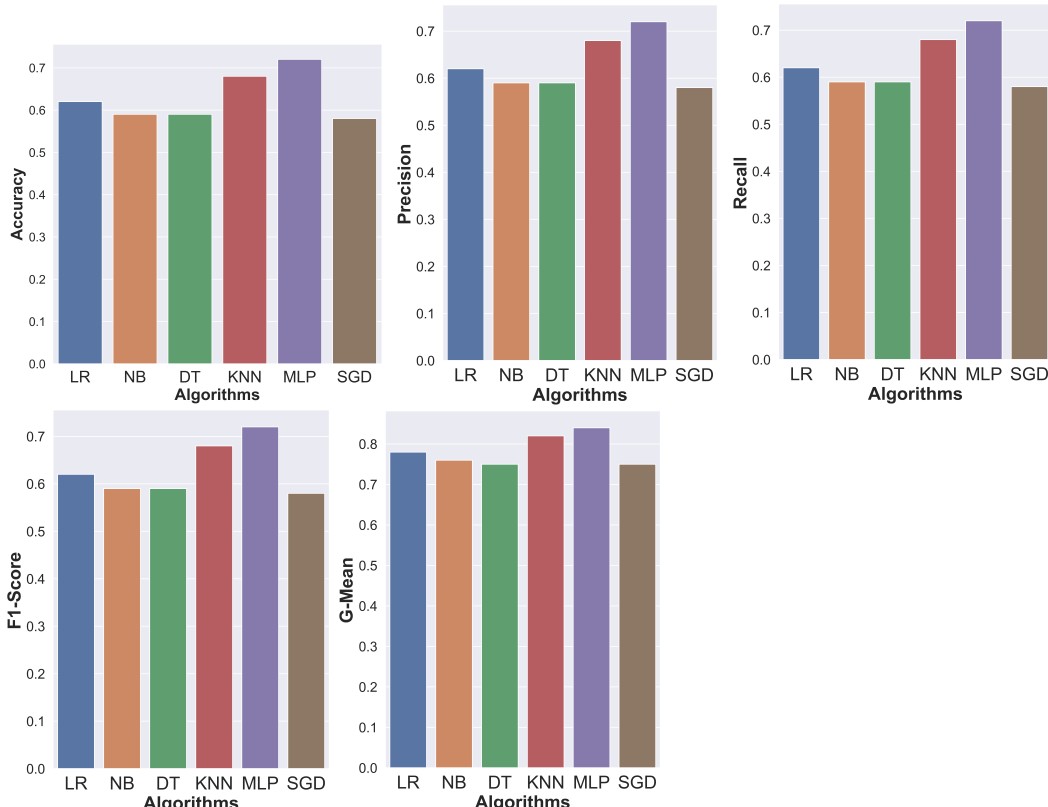

**Figure 8.** Score values of the various measures for BDD100K.

**Table 4.** Score values of the various measures for groups in the BDD100K dataset.

| Groups | Algorithms | Accuracy | Precision | F1-Score | G-Mean | Recall |
|---|---|---|---|---|---|---|
| Vehicle group | DT | 89% | 89% | 89% | 93% | 89% |
| | NB | 94% | 94% | 94% | 96% | 94% |
| | KNN | 94% | 94% | 94% | 96% | 94% |
| | MLP | 94% | 94% | 94% | 96% | 94% |
| | SGD | 93% | 93% | 93% | 95% | 93% |
| | LR | 94% | 94% | 94% | 96% | 94% |
| People group | DT | 91% | 91% | 91% | 91% | 91% |
| | NB | 95% | 95% | 95% | 95% | 95% |
| | KNN | 95% | 95% | 95% | 95% | 95% |
| | MLP | 95% | 95% | 95% | 95% | 95% |
| | SGD | 93% | 93% | 93% | 93% | 93% |
| | LR | 95% | 95% | 95% | 95% | 95% |
| Sign group | DT | 68% | 68% | 68% | 68% | 68% |
| | NB | 57% | 57% | 57% | 57% | 57% |
| | KNN | 73% | 73% | 73% | 73% | 73% |
| | MLP | 78% | 78% | 78% | 78% | 78% |
| | SGD | 71% | 71% | 71% | 71% | 71% |
| | LR | 77% | 77% | 77% | 77% | 77% |

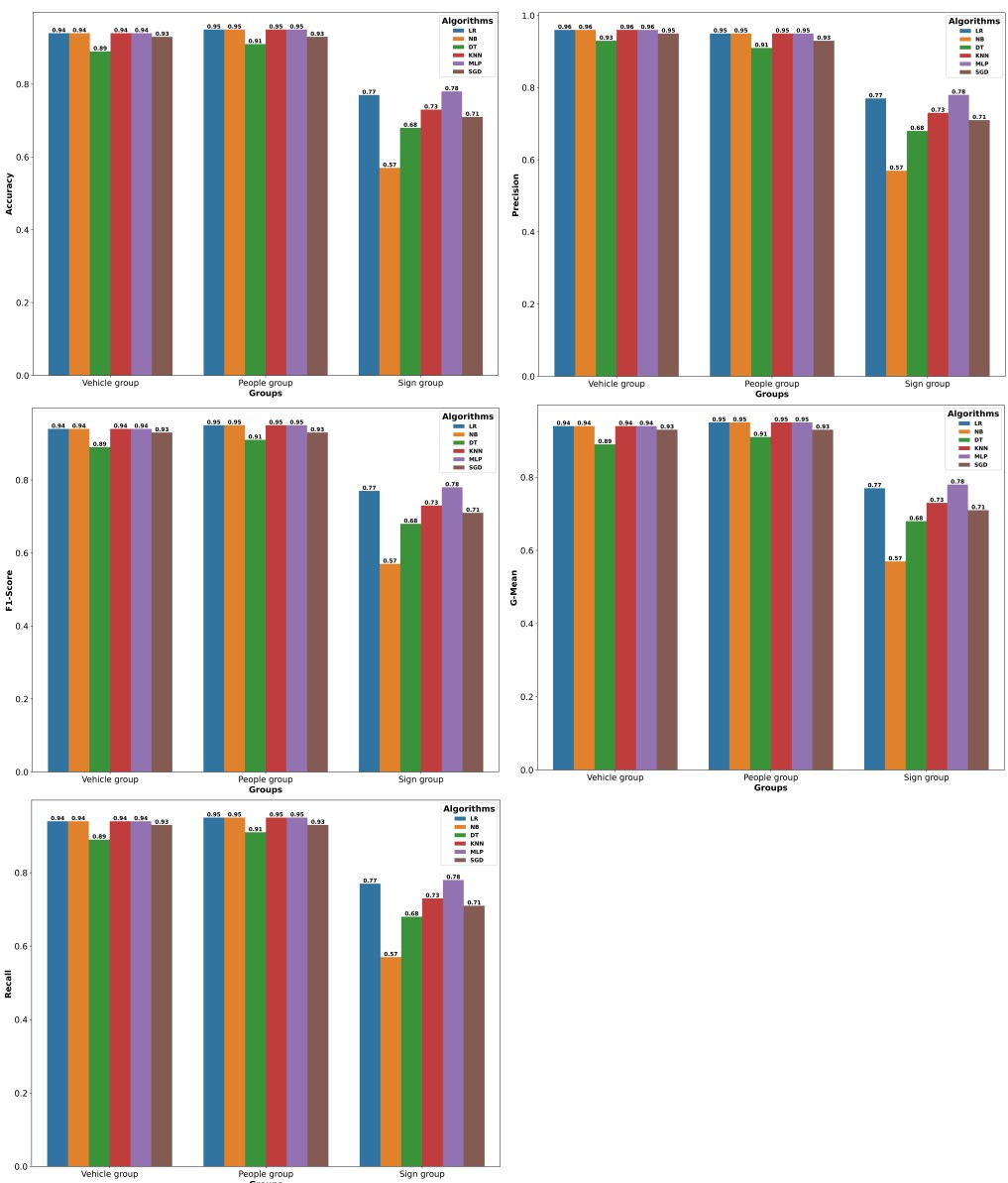

**Figure 9.** Score values of the various measures for groups.

**Table 5.** Comparison with similar work for the different measures (Udacity dataset).

| Measure | DT (C. R. Kumar, 2021) | DT (Our Approach) |
| --- | --- | --- |
| Accuracy | 93% | 97% |
| Precision | 90% | 98% |
| F1-score | 90% | 97% |
| G-Mean | X% | 97% |
| Recall | 90% | 97% |

Further, the test ranking for the Udacity dataset for the various classifiers is applied in Table 6. It can be observed from the table that the DT has achieved the best rank for all the measures followed by KNN. In contrast, NB has the worst ranking compared to the other classifiers. The Friedman chi-square value based on the ranks equals 25.0 with a *p*-value of 0.000139. Considering $\alpha = 0.05$, we can reject the null hypothesis indicating a significant difference between the various classifiers.

The second comparison is between the overall classes of the BDD100k dataset and the scores obtained after modifying the dataset by grouping similar categories into three main groups: vehicles, people, and signs. We got results that we compared with results obtained from the grouped dataset. After grouping the dataset, we saw an improvement in all of the measures adopted in our study. The following are our findings for the various measures we used: the MLP algorithm yielded the best results for G-Mean in BDD100K with an 84% score. In contrast, the best result we obtained from the group dataset of vehicles using the LR, NB, and KNN for G-Mean was 96%. The LR, NB, KNN, and MLP algorithms achieved the most outstanding results on all scales for 90% of the people group. The MLP algorithm achieved 78% on all measures for the group of signs, the highest score in the group. Except for the NB in the group of signs, all results in split groups were better than the overall results of the dataset. Here, we can state that collecting the original data into groups improved the prediction and classification results. This is the objective of our work by enhancing the performance of detecting the existing objects from the dataset used.

**Table 6.** Ranks for the different measures (Udacity dataset).

|  | DT | NB | KNN | MLP | SGD | LR |
|---|---|---|---|---|---|---|
| Accuracy | 1 | 6 | 2 | 3 | 4 | 5 |
| Precision | 1 | 6 | 2 | 3 | 4 | 5 |
| F1-Score | 1 | 6 | 2 | 3 | 4 | 5 |
| G-Mean | 1 | 6 | 2 | 3 | 4 | 5 |
| Recall | 1 | 6 | 2 | 3 | 4 | 5 |
| Total Rank | 5 | 30 | 10 | 15 | 20 | 25 |

## 6. Conclusions and Future Work

This paper shows a comparison of machine learning algorithms. Six supervised learning algorithms are compared in this study. The datasets used in this research are Udacity and BDD100K. In the training step, we use 80% of the dataset to train each algorithm. Finally, methods are used to evaluate 20% of the datasets. The results show the decision tree we used produced a G-Mean of 98%, with the highest score in the Udacity dataset and the best score across all experiments. In contrast, the results obtained from the groups were better than the original dataset for the vehicles group using the LR, NB, and KNN with a G-Mean of 96%. The results of the algorithms were improved due to dividing the original dataset into groups, which, in turn, accomplishes the objective of our research, which is the increase in performance using our approach. The vehicle groupings come out on top with the highest score possible for this measure. The results for the signs group were the worst of all of them. However, except for the Precision measure, the remaining measures for the people group had the best results for all the groups. In the future, other studies might consider data classification according to the object's size, speed, and whether it is moving or fixed. In addition, experiments can be conducted with other algorithms.

**Author Contributions:** Conceptualization, M.B.Y., and R.Q.; methodology, M.B.Y.; software, M.A.; validation, M.A., R.Q. and M.B.Y.; formal analysis, M.A.; investigation, M.A. and M.B.Y.; resources, M.A., R.Q. and M.B.Y.; data curation, M.A., R.Q. and M.B.Y.; writing—original draft preparation, M.A.; writing—review and editing, M.B.Y.; visualization, R.Q.; supervision, R.Q. and M.B.Y.; project administration, M.A., R.Q. and M.B.Y.; funding acquisition, M.B.Y. All authors have read and agreed to the published version of the manuscript.

**Funding:** This research received no external funding.

**Data Availability Statement:** Data will be available to any reasonable request.

**Conflicts of Interest:** The authors declare no conflict of interest.

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
