# Peer review of "An Object Classification Approach for Autonomous Vehicles Using Machine Learning Techniques"

_wevj, doi:10.3390/wevj14020041_

Round 1

Reviewer 1 Report

"This study compares several ML tools for object classification mechanisms for autonomous vehicles. I believe the manuscript needs some major revisions to be suitable for publication."

First, the contribution should be presented more clearly to show this comparison's importance and position in the literature, especially with other comparison articles.

Second, the review of different ML tools should be illustrated with some insightful depth giving the most important mathematical models.

Third, all the experimental studies should present more fancy statistical analyses to show the true differences among the compared methods.

“Fourth, the authors should shed light on the correlated problem of vehicle sensor location problem as in the following studies: 1.Distributing portable excess speed detectors in AL riyadh city, 2. Exact and Heuristics Algorithms for Screen Line Problem in Large Size Networks: Shortest Path-Based Column Generation Approach, 3. Traffic sensor location problem: Three decades of research , 4. A factorization scheme for observability analysis in transportation networks, 5. Location strategy for traffic emission remote sensing monitors to capture the violated emissions, 6. Robust deep learning architecture for traffic flow estimation from a subset of link sensors

Author Response

First Reviewer:

"This study compares several ML tools for object classification mechanisms for autonomous vehicles. I believe the manuscript needs some major revisions to be suitable for publication."

Response: We thank the reviewer for the time and effort given to read the paper. The given comments have helped us to improve the submitted manuscript. 

First, the contribution should be presented more clearly to show this comparison's importance and position in the literature, especially with other comparison articles.

Response: We thank the reviewer for raising this question regarding the importance of the proposed work and how different it is from previous studies. First, selecting a certain machine learning to help autonomous vehicles to detect and classify objects over the road networks is an important topic. The more suitable the selected machine, the more accurate and efficient the detection process. Thus, the safer and more secure the trip of the autonomous vehicle. This study has compared the performance of several suggested machine-learning mechanisms using different datasets. Second, several previous studies in this field of research have failed to recognize small objects, while the other ones failed to identify large ones. In adverse weather circumstances, a number of previous studies have produced poor results. Moreover, previous studies have considered one dataset and investigated the performance of fewer machine learning mechanisms. In our proposed work, these issues have been handled and accurate predictions have been obtained for both large and small objects and under all weather conditions. We added this clarification directly to the introduction.

Second, the review of different ML tools should be illustrated with some insightful depth giving the most important mathematical models.

Response: Following the reviewer's recommendation, we added more mathematical details to the considered machine learning tools.  Please check the background Section in the paper.

Third, all the experimental studies should present more fancy statistical analyses to show the true differences among the compared methods.

Response: We tried to introduce more statistical analysis to the obtained results following the recommendations of the reviewer.

 “Fourth, the authors should shed light on the correlated problem of vehicle sensor location problem as in the following studies:

1.Distributing portable excess speed detectors in AL riyadh city,

  1. Exact and Heuristics Algorithms for Screen Line Problem in Large Size Networks: Shortest Path-Based Column Generation Approach,
  2. Traffic sensor location problem: Three decades of research,
  3. A factorization scheme for observability analysis in transportation networks,
  4. Location strategy for traffic emission remote sensing monitors to capture the violated emissions,
  5. Robust deep learning architecture for traffic flow estimation from a subset of link sensors”

Response:

Thank you for this note that aims to strengthen and improve the research. We have referred to these papers that considers the vehicle sensor location in this work.  We plan to include this consideration in our future research and study its importance in regulating the movement and speed of traffic and errors in sensor measurements, and it should provide more accurate estimates of traffic flow. For now, we consider a dataset that has been collected at a constant speed and fixed distances of sensors.

Reviewer 2 Report

This research presents a good technique to improve the performance of classification algorithm by data processing method, different supervised learning algorithms are compared. and it has been validated on the datasets with high practicality. It is a smart object classification approach for autonomous vehicles, and it provides an efficient model so that former research’s weaknesses can be addressed.

There are still some minor points can be improved, e.g.

Figure 1 should be Table 1 according to the style of the content.

Some Figures’ resolution can be improved, e.g., Figure 3, 10.

It will be better If the classification algorithm considers size, velocity of the target objects.

Author Response

Second Reviewer:

This research presents a good technique to improve the performance of classification algorithm by data processing method, different supervised learning algorithms are compared. and it has been validated on the datasets with high practicality. It is a smart object classification approach for autonomous vehicles, and it provides an efficient model so that former research’s weaknesses can be addressed.

Response: We thank the reviewer for the positive appreciated feedback

There are still some minor points that can be improved, e.g.

Figure 1 should be Table 1 according to the style of the content.

Response: We fixed this in Table 1

Some Figures’ resolution can be improved, e.g., Figure 3, 10.

Response: We enhanced the performance of the used figures.

It will be better If the classification algorithm considers size, velocity of the target objects.

Response: In this work, we utilize some popular and highly used datasets that did not consider the velocity or mobility of objects. However, the size of each object can be represented as the width, height, or area of the object.  These parameters can be used as input features for the classification algorithms. This allows the algorithm to consider the relative size of different objects and make more informed decisions about their class. Indeed, classification based on size has been tested in our experiments, but classifying objects according to their nature yielded better results so we apply the grouping process based on this factor. Regarding the velocity, it should be a good experiment to use the movement of the object in consecutive frames. This can be represented as the change in position, width, height, or other relevant features between the frames. This information can be used as an additional feature in the classification algorithm, allowing it to consider the movement of the object and make more informed decisions about its class. However, the collected images were taken at a constant speed in the used dataset and it did not consider this parameter. We added this feature to our future studies section to focus on classifying objects using images taken at various speeds.

Reviewer 3 Report

Please find the comments referring to the paper as an attachment.

Author Response

Third Reviewer:

The paper is properly developed in terms of content and editorial. It concerns important topics and artificial intelligence tools (neural network) were used for research. The accuracy of the obtained results is presented. However, this paper requires improvements, as detailed below.

Response: We thank the reviewer for the positive appreciated feedback.

 Detailed comments:

  1. The first part of the abstract is typical of the Introduction section. Abstract should be a short guide to the article with an indication of its new achievements.

Response: we fixed the abstract and made it shorter directly indicating the achievements.

  1. Introduction section: Literature references should be written in the form 2‐4, 5‐8, etc. In addition, the content of the cited articles 2‐11 should be summarized. This section should also present the possibilities of theoretical and/or practical application of the discussed topic.

Response: We fixed the references as recommended. We make sure to summarize the main contribution of each work. The possibilities of practical applications of object detection using machine learning techniques have also been discussed in the introduction.

  1. Please write the product sign using a dot, not an asterisk, e.g. Eq. 1.

Response: Done.

Reviewer 4 Report

This manuscript wanted to present an object classification approach for autonomous vehicles. However,  I cannot find any new things from the manuscript.

(1) The mentioned methods, i.e., Decision Tree (DT), Naive Bayes (NB), K-Nearest Neighbor (KNN), Stochastic Gradient Descent (SGD), Multi-Layer perceptron(MLP), and Logistic Regression (LR) algorithms, are very old, and no any improvements were done these methods. What are the main work done by the authors?

(2) The literature review is very selected. A lot of work on object detection and classification in CV have been done. The authors should listed the main and important work in this section.

(3) The deep learning based methods have dominated the object detection and classification in CV, why the authors chose the outdated methods for autonomous vehicles? 

Based on the discussion above, I suggest the authors should conduct more innovative work before they submit the paper.

Author Response

Fourth Reviewer:

This manuscript wanted to present an object classification approach for autonomous vehicles. However, I cannot find any new things from the manuscript.

Response: We thank the reviewer for the time and effort given to read and evaluate our paper. We tried our best to follow the suggestions and clarify the work 

  • The mentioned methods, i.e., Decision Tree (DT), Naive Bayes (NB), K-Nearest Neighbor (KNN), Stochastic Gradient Descent (SGD), Multi-Layer perceptron (MLP), and Logistic Regression (LR) algorithms, are very old, and no any improvements were done these methods. What are the main work done by the authors?

Response: We did not make improvements to these techniques since our contribution has focused on enhancing the datasets themselves. Based on the results of several experiments conducted when the Udacity dataset classes were separated, we ultimately decided to group classes that were relatively similar into a single class. As it has a record of very few observations and also because it belongs to the same category, the decision to gather it was a decision that aimed to improve the results for all tested machine-learning techniques.  Moreover, the case in the second dataset BDD100K, where the dataset was classified into groups based on the nature of the objects. The performance of the used algorithms has been improved in comparison with the original dataset. The main contribution of this work is a smart object classification approach for autonomous vehicles, and it provides efficient results.

  • The literature review is very selected. A lot of work on object detection and classification in CV have been done. The authors should list the main and important work in this section.

Response: We aimed to investigate the most relevant and recent studies of our work. The main challenges and drawbacks of each previous study have been investigated.

  • The deep learning-based methods have dominated the object detection and classification in CV, why the authors chose the outdated methods for autonomous vehicles? 

Response: Adopting these techniques was not a matter of chance in our investigation; rather, it was the result of trial and error. The nature of the approach we used to demonstrate our contribution is largely responsible for the fact that the results we produced utilizing the aforementioned six algorithms were the best. We definitely plan to investigate deep learning and neural network technologies in future studies.

Based on the discussion above, I suggest the authors should conduct more innovative work before they submit the paper.

Response: We hope the new version of the paper will come to your satisfaction.

Round 2

Reviewer 1 Report

None.

Author Response

Thank you for the positive comments

Reviewer 4 Report

There are still some problems in the manuscript.

(1) After reading the response of the authors, I still cannot get what are the main contributions of the study? The authors should state them in the manuscript clearly.

(2) The authors claimed that their main contributions were not the improvements of the presented six ML methods and it is datasets enhancement. If so, the whole logic of the manuscript should be reformulated according this research objective, and the literature review should be changed.

(3) From the sections abstract, introduction the literature review, it is difficult to conclude the research problems.

(4) The title of the paper is very common and it should be changed to a more suitable one.

Author Response

We thank the reviewer for the valuable comments we provide detail responses to the comments here:

  • After reading the response of the authors, I still cannot get what are the main contributions of the study? The authors should state them in the manuscript clearly.

Response: This work aims to develop an intelligent object classification mechanism for autonomous vehicles. The proposed mechanism uses machine learning technology to predict the existence of investigated objects over the road network early. We use different datasets to evaluate the performance of the proposed mechanism. Accuracy, Precision, F1-Score, G-Mean, and Recall are the measures considered in the experiments. Moreover, the proposed object classification mechanism is compared to other selected previous techniques in this field. The results show that grouping the dataset based on their mobility nature before applying the classification task improved the results for most of the algorithms, especially for vehicle detection.

We clearly added this to the abstract and the introduction section.

  • The authors claimed that their main contributions were not the improvements of the presented six ML methods and it is datasets enhancement. If so, the whole logic of the manuscript should be reformulated according this research objective, and the literature review should be changed.

Response: We compare the performance of six main machine learning to recommend the most suitable one for object detection on road networks. The performance of the proposed approach has been improved by  modifying the datasets and group similar objects in a single group.

We clearly added this target to the paper.

  • From the sections abstract, introduction the literature review, it is difficult to conclude the research problems.

Response: Detecting and classifying the objects over the road networks to allow drivers to timely respond to their existence is the main problem that has been handled in this paper.

 we tried to clarify it in the last version of the paper.

  • The title of the paper is very common and it should be changed to a more suitable one.

Response: We changed the title